# Database and AI Diagnostic Tools Improve Understanding of Lung Damage, Correlation of Pulmonary Disease and Brain Damage in COVID-19

**DOI:** 10.3390/s22166312

**Published:** 2022-08-22

**Authors:** Ilona Karpiel, Ana Starcevic, Mirella Urzeniczok

**Affiliations:** 1Łukasiewicz Research Network—Institute of Medical Technology and Equipment, 41-800 Zabrze, Poland; 2Laboratory for Multimodal Neuroimaging, Institute of Anatomy, Medical Faculty, University of Belgrade, 11000 Belgrade, Serbia

**Keywords:** artificial intelligence, databases, lung diseases, EEG, brain damage, AI diagnostic, pulmonary disease, SARS-CoV-2

## Abstract

The COVID-19 pandemic caused a sharp increase in the interest in artificial intelligence (AI) as a tool supporting the work of doctors in difficult conditions and providing early detection of the implications of the disease. Recent studies have shown that AI has been successfully applied in the healthcare sector. The objective of this paper is to perform a systematic review to summarize the electroencephalogram (EEG) findings in patients with coronavirus disease (COVID-19) and databases and tools used in artificial intelligence algorithms, supporting the diagnosis and correlation between lung disease and brain damage, and lung damage. Available search tools containing scientific publications, such as PubMed and Google Scholar, were comprehensively evaluated and searched with open databases and tools used in AI algorithms. This work aimed to collect papers from the period of January 2019–May 2022 including in their resources the database from which data necessary for further development of algorithms supporting the diagnosis of the respiratory system can be downloaded and the correlation between lung disease and brain damage can be evaluated. The 10 articles which show the most interesting AI algorithms, trained by using open databases and associated with lung diseases, were included for review with 12 articles related to EEGs, which have/or may be related with lung diseases.

## 1. Introduction

Respiratory diseases are one of the leading causes of morbidity and mortality worldwide. This situation results from systematic aging of the population, the prevalence of smoking and exposure to air pollution. By May 2022, there was a sharp increase in interest in both machine learning and the application of artificial intelligence, in particular as a tool supporting the diagnosis of lung diseases. The COVID-19 pandemic had a direct impact on this phenomenon. The term “pneumonia” appears in the PubMed database search engine in the period of January 2019–May 2022 over 150,000 times. Narrowing the search to “pneumonia” and “chest X-ray”, we have over 3000 results, while “detecting pneumonia” and “artificial intelligence”, in the same range, gives 689 results. The searches have been narrowed down to the works that indicate direct access to databases.

Artificial intelligence finds applications not only in aiding pneumonia detection but also in: chest screening, sensing lung nodules, fibrosis, effusion, mass, cardiomegaly, cardiac hypertrophy, pulmonary edema, opacity or pleural effusion. A digital chest X-ray is one of the most common imaging examinations.

Hospitals collect a very large amount of data, in particular X-ray images along with radiological reports. In addition to X-ray images, a large number of MRI and CT examinations are performed in the imaging diagnostics facility, which require a large amount of available space. The data is stored in picture archiving and communication systems (PACS) imaging systems. As of today, there are many questions that we cannot answer, in particular how to use data in building precise computer-aided diagnostics (CAD) systems. Another issue is how to obtain data from more medical facilities so that the learning process is even more precise. So far, most of the publications are based on a certain number of shared photo/study databases. Therefore, we have access to data, which allows us to create more and more accurate algorithms supporting diagnostics in almost every field.

Today, we know the consequences of the COVID-19 pandemic. Complications that occur during or immediately after the disease are very dangerous. Based on the latest data on the pathogenesis of prion diseases and the immune response to SARS-CoV-2, it has been hypothesized that a cascade of systemic inflammatory mediators in response to the virus accelerated the pathogenesis of prion disease [1]. This means that, along with the diagnosis of lung disease, we should look at the central nervous system (CNS) and we need to evaluate the correlation with specific brain morphological substrates. Therefore, it is extremely important to develop equally fast and precise algorithms that will support the diagnosis of the above mentioned possible correlated brain structures with lung issues. For starters, the most common method is EEG testing, as one of the oldest and still most reliable neuroimaging techniques for identifying electric activity of the brain and its specific structures. It turns out that this topic is not fully exhausted yet, and there are not many EEG databases that contain specific pathologies or diseases, in particular, considering the EEG tests of patients after COVID-19, who experience various neurological problems and require rehabilitation. The correlation between the occurrence of brain damage after COVID-19 has already been noticed and described in the literature [2,3,4]. The topic, however, seems to still be new and more observations and research are needed to establish the correlation ratio [5,6]. This makes it all the more reasonable to create overviews that allow researchers to find as much data as possible without spending a lot of time.

## 2. Machine Learning Tools

### 2.1. Tools, Libraries and Blogs

Based on the data available in Google Trends, it is clear that interest in the keyword “machine learning” is growing, especially since the beginning of 2015. Noticeably, the most used language is R and Python. Libraries, packages most often used for calculations, include: NumPy, Pandas and SciPy. The following are used for data visualization: Seaborn (https://seaborn.pydata.org/introduction.html, accessed on 5 July 2022), Bokeh (https://docs.bokeh.org/en/latest/, accessed on 5 July 2022), Plotly and Matplotib (https://matplotlib.org/2.0.2/users/pyplot_tutorial.html, accessed on 5 July 2022). However, the most well-known packages are: Keras, Sci-Kit Learn (https://scikit-learn.org/stable/auto_examples/index.html, accessed on 5 July 2022), Theano, NLTK, Gensim, PyLearn2, Lasagne, Caffe, Torch7, Deeplearning4j and Tensorflow (https://www.tensorflow.org/guide, accessed on 5 July 2022). Statistical support libraries are also worth mentioning, such as Scrapy or Statsmodels. These are not all the available libraries, but only the most popular (https://numpy.org/arraycomputing/, accessed on 5 July 2022). A total of 51 most used machine learning tools by experts were described in techvidvan [7,8]. NumPy was introduced in 2006, Pandas appeared on the landscape in 2008. Many newer libraries mimic NumPy-like features and capabilities and pack newer algorithms and features geared towards machine learning and artificial intelligence applications. If we are looking for libraries for other programming languages, a very good overview is provided in GitHub Awesome Machine Learning (https://github.com/josephmisiti/awesome-machine-learning, accessed on 5 July 2022). A fair number of researchers use the Kaggle platform, which has a considerable amount of data and tutorials. In one place, we can find machine learning algorithms for use in data science projects (https://www.kaggle.com/code/shivamb/data-science-glossary-on-kaggle/notebook, accessed on 5 July 2022). It is also noteworthy to have access to a free and open source resource with machine learning papers with code and evaluation tables. Anyone can join and add their implementation to a given paper (https://paperswithcode.com/sota, accessed on 5 July 2022). The mission of the service is to present the latest advances in the field of machine learning. Data visualization is also an important aspect. Visual Capitalist supports making data visualization as simple as possible (https://www.visualcapitalist.com/, accessed on 5 July 2022).

Not every user has the right hardware, graphics card and software. Python is an accessible and free programming language, which can be easily implemented in free interactive environments. Such environments include, for example, colab (https://colab.research.google.com, accessed on 5 July 2022), Kaggle, or Anaconda. Currently, the most effective neural network architectures used for image processing can be listed: AlexNet [9], VGGNet [10], Xception [11], ResNet [12] and DenseNet [13]. These are learned models supporting the feature extraction procedure, which are based on convolutional layers. Of course, the number of packages and libraries that are worthy of interest will grow over time and we can only guess what else will appear. It is certainly not the case to always use all of them. Everything depends on the project and solution. 

It is not just the tools themselves that have a direct impact on the creation and development of ever newer algorithms. Towards Data Science is a platform that features articles on data science, machine learning, visualization and programming. They collaborate with more than ten editorial boards, making it a great resource (https://towardsdatascience.com/, accessed on 5 July 2022). There are also a lot of popular science blogs, where you can find a lot of inspiring articles that make it easier to explore machine learning. A blog worth noting is one by a true master of machine learning, Jason Brownlee (https://machinelearningmastery.com/blog/, accessed on 5 July 2022). Brownlee posts articles simply explaining how machine learning algorithms work and more advanced ones supporting the work of experienced researchers. Becoming Human is another blog with information and tutorials on artificial intelligence and machine learning. In addition, you can find information about the latest developments in AI and the benefits of artificial intelligence development for humans (https://becominghuman.ai/, accessed on 5 July 2022). You can also browse 90 of the most popular blogs on the topic discussed (https://blog.feedspot.com/data_science_blogs/, accessed on 5 July 2022) and if you are interested in the latest news and updates in artificial intelligence, this information can be found on the AWF Machine Learning Blog (https://aws.amazon.com/blogs/machine-learning/, accessed on 5 July 2022).

The researcher may also use Open AI, a blog through which you can access research papers on artificial intelligence. This blog focuses on long-term research where research papers are always available to the general public (https://openai.com/blog/, accessed on 5 July 2022). Tools have been developed to support and facilitate analysis such as Data Is Beautiful (https://www.reddit.com/r/dataisbeautiful/, accessed on 5 July 2022). The site offers unique ideas for presenting some data. 

Analytics Vidhya has a lot of educational material on artificial intelligence, machine learning and deep learning. The team presents detailed and high-quality tutorials on topics related to neural networks (https://www.analyticsvidhya.com/blog/, accessed on 5 July 2022).

In summary, there are more and more tools, libraries and blogs available for use. They are contributing to a growing understanding of artificial intelligence issues. Most importantly, the methods are becoming more accessible and the materials available allow for rapid user implementation. More and more researchers can improve existing algorithms, create their own and implement cutting-edge solutions.

### 2.2. Open Databases

Machine Learning is not possible without data that contains the information we need, allows us to ask questions and enables us to find valuable answers that will turn into conclusions. Databases used in machine learning algorithms to diagnose lung diseases are now increasingly available and free of charge. Below shows selected the set of datasets:

For lung diseases:OPENI NLM NIH 4 May 2022 [14]NIH chest X-rays image dataset 4 May 2022 [15]The PLCO dataset 4 May 2022 [16]The MIMIC-CXR database 4 May 2022 [17]COVID-19 Image Data Collection 4 May 2022 [18]The Korean Institute of Tuberculosis dataset 4 May 2022 [19]CheXpert: A Large Chest Radiograph Dataset with Uncertainty Labels and Expert Comparison 4 May 2022 [20,21]The Indiana University dataset 4 May 2022 [22,23,24]The JSRT dataset 4 May 2022 [25]Actualmed COVID-19 Chest X-ray Dataset Initiative 4 May 2022 [26]COVID-19 Radiography Database 4 May 2022 [27]RSNA Pneumonia Detection Challenge 4 May 2022 [28]COVID-Net Open Source Initiative 4 May 2022 [29]Masks of the lung area for the evaluation of segmentation performance 4 May 2022 [30]The Shenzhen dataset 4 May 2022 [31]Lung Image Database Consortium (LIDC) 4 May 2022 [32]Biobank 26 July 2022 [33]

For the central nervous system:OSF HOME 4 May 2022 [34]Physionet, list of all the databases 4 May 2022 [35]A list of all public EEG datasets 4 May 2022 [36]UCI EEG Database 4 May 2022 [37]ENGINEURING 4 May 2022 [38]ADNI 4 May 2022 [39]HeadIT 4 May 2022 [40]SCCN, EEG/ ERP 4 May 2022 [41]BRAIN SIGNALS 4 May 2022 [42]OPEN NEURO 4 May 2022 [43]BNCI–HORIZON [44]EEG Dataset, I, II, III 4 May 2022 [45]MAMEM Phase 4 May 2022 [46]The Patient Repository for EEG Data 4 May 2022 [47]EEG with positive test PCR 25 July 2022 [48]

These databases are a good source that users can use to create a model of their choice, based on machine learning. It is worth mentioning that there is a growing need to share more and more data and to improve existing algorithms and create new ones. The existing overview should be successively updated so that users, developers and researchers who create algorithms have up-to-date databases. 

## 3. Systems and Applications

Diagnostic physicians and radiologic technicians must rely primarily on their knowledge and experience when analyzing X-ray images. In most cases, diagnostic physicians use available applications to support the process of analyzing X-ray images, these applications usually offer modest capabilities for modifying a given image, such as changing brightness, negation, etc. (eFilm, Onis, Osiris, Alteris). In many cases, simple methods are not sufficient, as many lesions are not visible after using simple image processing, by this it is not possible to unambiguously indicate whether an object on an X-ray image is a lesion. Computerized methods for analyzing and processing digital images are a solution to this problem. Thanks to the use of IT methods, it becomes possible to extract the lesions of interest to the diagnostician. This makes cooperation on the line between medicine and informatics even more necessary.

Currently, algorithms and methods of computerized processing of medical images are mainly used for computed tomography, the average waiting time for a chest CT scan is about 100 days, but also the greater harmfulness of this type of examination determines the very frequent use of X-ray machines in early medical diagnosis Section 3.1.

### 3.1. BlueDot, Application of AI to COVID-19

BlueDot, Tools to Help Predict Pandemics and the Spread of Diseases (demonstrated by a Canadian company), uses artificial intelligence to scour the internet for signals indicating epidemiological risk [49]. The system builds complex models of disease occurrence and spread based on data from more than 100 sources. These include local news, online forums, data from hospitals and information on illnesses among animals. The algorithm even takes into account demographic data, local and air transportation or climate data.

Companies like BlueDot use a range of Natural Language Processing (NLP) algorithms to monitor news sites and official health care reports in different languages around the world, marking whether they mention high priority diseases such as COVID-19 or others such as HIV/AIDS or tuberculosis. BlueDot, on the other hand, was able to predict the COVID-19 epidemic and warn its users even before the World Health Organization did [50]. 

Ultimately, the biological aspect of the coronavirus could not be investigated as data were lacking. Without them, artificial intelligence is useless. For this reason, it has been impossible to detect a completely new virus such as the coronavirus responsible for COVID-19.

### 3.2. Existing Methods for Pneumonia Brain Correlation Detection

There are data showing a strong correlation between brain-related abnormalities and COVID-19 [51,52,53]. However, it is still not quite certain that, except for the mild and severe cases, early brain damage might be caught in order to reveal possible pathophysiological mechanisms contributing to brain pathology and its correlation with lung issues [54].

Therefore, we would like to emphasize the importance of database and AI involvement in the early detection of the lung–brain correlation in COVID-19 cases. The availability of pre-infection imaging data would and will reduce the possibility of damaging risk factors being misinterpreted as COVID-19 effects and would also pinpoint the targeted involved specific brain structures [55,56,57,58,59,60]. Importantly, emerging literature is showing that the virus may spread to the central nervous system through neuronal routes, hitting the brainstem and cardiorespiratory centers, potentially exacerbating the respiratory illness [61]. The lung disease severity score may be predictive of acute abnormalities on neuroimaging in patients with COVID-19 with neurologic manifestations. This can be used as a predictive tool in patient management to improve clinical outcome [5].

It should be noted that due to various divergent pathologies and potential specialist fatigue, errors in the interpretation of medical images may occur [62]. The time of the pandemic has influenced the development and emergence of new solutions and technologies. The first CNN models appeared at the beginning of the pandemic, but they were based on a limited number of chest X-rays and were characterized by unsatisfactory performance. Recently, there has been greater interest in this subject, which translates into an increase in the number of solutions, methods and tools based on more and more advanced algorithms [63,64,65,66,67,68,69], that, on the one hand, makes it difficult to make a selection in search of a good/best solution and, on the other hand, we have a large number of already ready-made solutions [70,71,72,73,74,75,76,77]. The theory is quite applicable in practice, as in 2022 there are several tools that can be used while being at home (for example, CheXNet [78], UBNet [79], PneumoniaNet [80]), which has allowed for the rapid development of telemedicine and patient–doctor contact at a distance. In Poland, one of the first to appear was the so-called CIRCA [81]. It uses machine learning techniques that allow doctors on duty in emergency rooms and hospital wards to make an initial assessment of the nature of changes in the lung region of patients with respiratory disorders. The system enables the identification of patients requiring different supplies from medical personnel. It also facilitates diagnosis in the case of a large number of infections, providing the possibility of separating, on the basis of a generally available X-ray, a group of people at high risk of imaging changes typical of COVID-19.

The least common form, but also very rapidly developing, is the analysis of data (X-ray images) taken with a smartphone or using a smartphone. Now, in 2022, we have datasets derived from two large and publicly accessible digital CXR databases [82] (MIMIC-CXR [21] and CheXpert [17,21]). Blending Artificial Intelligence (AI) with chest X-ray images and incorporating these models in a smartphone can be handy for the accelerated diagnosis of COVID-19. In the last 3 years, several applications have appeared that are still evolving, i.e., Pneumonia Detection [83] or Lung Cancer Detection for Android [84], XraySetu [85], Chest X-ray Interpretation [86] and CheXphoto [87]. 

The use of artificial intelligence has as many supporters as opponents, but it has been proven that AI techniques with chest X-ray images can encourage specialists to complete a thorough analysis in a short time. Some regions have difficult access to specialists or hospital facilities, which makes early detection and diagnosis a difficult task. Smartphone [88,89,90], tablet or PC applications could in the future support the work of doctors in less accessible regions. The waiting time for an appointment, the cost of travel or the course of the diagnosis would definitely be shortened.

### 3.3. Chatbot, Application of AI

With regard to medical research, algorithms are also used to read diseases other than COVID-19. Chatbots can take over some of the doctor’s duties and shorten the waiting time for an appointment. Chatbots can be used, for example, when creating an interview with a patient or recognizing initial symptoms. Chatbots are already used to fight the pandemic, and have demonstrated high accuracy during screening for COVID-19 [91]. In addition to being used by patients, they have also been used in healthcare testing to detect COVID-19 and minimize virus transmission [92]. There is more and more talk about the wider use of chatbots, e.g., based on new components [93]. A chatbot called SGDormBot was used for mass screening of migrant workers in Singapore [94].

## 4. Results

PubMed, the Web of Science, ResearchGate and Google Scholar were searched for the relevant articles in the last three years with the search terms: “artificial intelligence applications”, “artificial intelligence tools”, “deep learning”, “neural network”, ”pulmonary nodules”, “lung cancer”, “respiratory medicine”, “lung changes”, “lung infection”, “pneumonia”, “COVID-19”. Inclusion criteria: (1) originality and innovation; (2) reliable source of data; (3) a good description of the data processing mechanism. Exclusion criteria: old and irrelevant literature. Literature was selected by two independent authors and read and discussed by all authors to extract useful information.

Based on above description, 10 articles were included for review (Table 1) associated with lung diseases and 12 were related to EEGs and cognitive decline, which is or may be related with lung diseases (Table 2). The selected collection represents access to the collections and a summary of the main results obtained by the methods presented. 

Among the all available online databases, the one used most often was created by Wang et al. [15,105]. This is not surprising considering it contains over 100,000 X-ray scans of over 30,000 unique patients. The available scans present lung diseases such as: pneumonia, edema, fibrosis, nodule mass and others. Each scan was tagged based on the radiologists’ opinion, which ensures high reliability (>90%). 

Over the past two years, there has been an emergence of many publicly available databases dedicated to COVID-19. The largest one for today is COVID-Net Open Source Initiative [29,101]. The database contains over 16,000 positive COVID-19 scans from over 2800 patients and it is constantly growing.

The biggest problem appears when searching for databases containing alternatives for X-ray imaging methods, such as magnetic resonance imaging (MRI) and computed tomography (CT). The only publicly accessible CT database found is the LIDC—The Lung Image Database Consortium [32]. It consists of diagnostic and lung cancer screening thoracic CT scans with marked-up annotated lesions. The database contains over 244,000 DICOM images. The image annotation process was performed by four experienced thoracic radiologists.

In 2022, still one of the most frequently described, analyzed and discussed respiratory diseases is COVID-19. Even mild forms of COVID-19 can present sustained neurocognitive deficits [117,118]. In this time, the relationship between neurodegenerative diseases and severe acute respiratory syndrome coronavirus 2 (SARS-CoV-2) is yet to be fully clarified [106]. Cognitive decline is observed before, during and after infection. Observed symptoms may include: Alzheimer disease (AD) (Holmes et al., 2009), Parkinson disease (PD) [119], multiple system atrophy [120], frontotemporal dementia (FD) [121], progressive supranuclear palsy, primary progressive aphasia and stroke. EEGs are one of the most common diagnostic methods used in neuroradiology. It is a complementary test that provides important information for diagnosis. In particular, EEGs can be used to assess encephalopathy, epileptogenicity and any focal abnormalities in patients with COVID-19. Some studies successfully described EEG findings in patients with COVID-19 [2,111].

## 5. Discussion

### 5.1. Diagnosis of Chronic Obstructive Pulmonary Disease

Even before COVID-19, on a smaller scale, they had begun to deal with algorithms that were trained to recognize various lung diseases. In particular, they were concerned with the detection of cancer, tumors or, for example, obstructive pulmonary disease. The prevalence of chronic obstructive pulmonary disease (COPD) or brain injuries was higher in COVID-19 patients with concomitant chronic conditions such as dyslipidemia, diabetes and hypertension kidney disease [122]. The algorithms that have been created, and the databases that have been collected, are a good source of more and newer algorithms that make it possible to predict not only COVID-19, but also to perform analyses in a wider range. It is worth pointing out that chronic obstructive pulmonary disease remains undiagnosed in many people. Hence, opportunities for the diagnosis of chronic obstructive pulmonary disease are being sought in other procedures. Lancet Digital Health has published a paper in which a neural network model was used to diagnose chronic obstructive pulmonary disease on the basis of images from low-emission computed tomography of the chest. To create three models of neural networks, data from the PanCan study were used, which concerned the possibility of screening lung cancer in people with a long history of smoking. Then, 2153 computed tomography scans obtained from the ECLIPSE study (observational study of patients with chronic obstructive pulmonary disease) were used for validation. The neural network with the best diagnostic value obtained an AUROC equal to 0.89. Using this network, the data from the ECLIPSE study yielded an AUROC of 0.99, a positive predictive value of 0.85 and a negative predictive value of 0.76 [123]. By applying this approach more widely, it could be possible to diagnose more chronic obstructive pulmonary disease in people who undergo a detailed pulmonary nodule diagnosis.

### 5.2. Brain Damage Complications Arising Directly or Indirectly from COVID-19 Pneumonia

The disease that has provided a new breed of coronavirus as severe as the severe acute respiratory syndrome coronavirus 2 is COVID-19. It has affected more than 170 mil people in more than 217 countries with a progressive and still ongoing increase in the number of people infected and inefficient diagnosis treatment. The consequence of all that is a new more efficient approach presented as deep learning/DL and machine learning/ML, assisting medical professionals in prompt and efficient approaches. In this review paper, we classified previous studies into specific uptown data AI approaches and techniques and databases/datasets established and extracted from previous studies aiming to achieve the possible prediction of COVID-19 and correlation of involved lung and brain damage [59,60].

It seems to be that one of the basic directions of ML development in healthcare will be the development of learning algorithms for subsequent disease entities. The system implementation process is not fast. 

It is now a common fact that SARS-CoV-2 infection can damage many organs other than the lungs. The most troubling and complicated is damage to the brain. Symptoms such as brain fog, fatigue and depression may be mild or pretty severe. Many studies and scientists implicate that treatment of those with long term brain injuries will strain the healthcare system for years to come. Understanding, defining the origin and treatment of COVID-19-related brain injury is a high priority for medical science. Because the studies evaluated patients who became sick with COVID-19 before vaccines were widely available, it is not clear if this issue damage happens among vaccinated people, therefore experts are hopeful vaccines would offer some protection against neurological damage, as they do help reduce the risk of other types of tissue damage. One of the first studies looked at more than 400 people aged between 51 and 81 who were positive for COVID-19 from the U.K. Biobank study. The MRI scans taken prior to infection were compared to those taken an average of five months after infection. COVID-19 brain-related abnormalities were found in a study that investigated brain changes in 785 participants of the U.K. Biobank study (aged 51–81 years), who were imaged twice using magnetic resonance imaging, including 401 cases who tested positive for infection with SARS-CoV-2 between their two scans with 141 days on average separating their diagnosis and the second scan, as well as 384 controls. The availability of pre-infection imaging data reduces the likelihood of pre-existing risk factors being misinterpreted as disease effects, which is very important. Their findings showed interesting results such as greater reduction in grey matter thickness and tissue contrast in the orbitofrontal cortex and parahippocampal gyrus, greater changes in markers of tissue damage in regions that are functionally connected to the primary olfactory cortex and a greater reduction in global brain size in the SARS-CoV-2 cases. The participants who were infected with SARS-CoV-2 also showed on average a greater cognitive decline between the two time points. Importantly, these imaging and cognitive longitudinal effects were still observed after excluding the 15 patients who had been hospitalized. These mainly limbic brain imaging results may be the in vivo hallmarks of a degenerative spread of the disease through olfactory pathways, of neuroinflammatory events or of the loss of sensory input due to anosmia. Whether this deleterious effect can be partially reversed, or whether these effects will persist in the long term, remains to be investigated with additional follow-up [52]. Researchers from the University of Oxford found that even people with mild COVID-19 symptoms had signs of slightly reduced brain size and subtle tissue damage, especially in the region of the brain associated with sense of smell. The fact that this study demonstrates a loss in brain volume over several months is concerning and could imply accelerated brain aging. Olfaction, as a sense of smell, presents a crucial indicator related to several feedback processes such as the unconscious response to the molecular sampling of the environment, which is very complicated, as along with the anatomical substrates that allow them. The pathway starts from highly specific odor receptors located on the roof of the nasal cavity. From that point, the stimuli converge in the olfactory bulb and through a multitude of projections toward the amygdala, septal nuclei, pre-pyriform cortex, entorhinal cortex, hippocampus and the subiculum, thalamus and the frontal cortex provide a unique dynamic system [124]. The olfactory bulb presents a neuroanatomical substrate constantly exposed to the external environment’s diverse impact, therefore, it is considered an immune organ that prevents the invasion of viruses into the CNS [52]. Its dysfunction may be considered as a predisposing factor to a worse treatment outcome in respiratory virus infections when this immunological function is impaired or disrupted as a result of aging or some pathological processes like SARS-CoV-2. Another investigation that studied 64 people, some of whom had been hospitalized with COVID-19, and others who had not been hospitalized but later experienced long-haul symptoms, showed the presence of damaged neurons and glial cells as fundamental cells in the brain. The study found evidence of brain inflammation that correlated with symptoms of anxiety reported by long-haul COVID-19 patients. It was reported that about a third of people with COVID-19 developed some form of long-COVID-19 symptoms and many of them were neurological and psychiatric symptoms such as decreased memory, headache and dizziness. It was predicted that COVID-19-related neurological symptoms could become even more prevalent in the decade to come [125]. As previously mentioned, cognitive and neurological symptoms are common in people with long or prolonged COVID-19, and the symptoms can be debilitating for those affected. While growing evidence suggests that the SARS-CoV-2 virus causes damage to the central nervous system, the underlying mechanisms are not well understood. There is a term, brain fog, which explains the impact that this virus induces on cognitive functions. Some researchers are using the condition “neuro-COVID Trusted Source” to describe this presentation of the disease [126,127,128]. An experimental study in primates reported that SARS-CoV-2 infection causes brain inflammation and even cell death, apoptosis among other forms of brain injury [129]. There are also studies showing parallels of the effect of SARS-CoV-2 infection on the brains of primates with studies carried out on human autopsies on human brains of people who had died from COVID-19, but the inability to distinguish between the damage caused specifically by the virus and other factors is a limitation of this research [130]. There has been growing interest in electroencephalographic (EEG) data mining related to COVID-19 aiming to define specific-features EEG of encephalopathy in COVID-19. EEGs were and are most commonly ordered for an altered level of consciousness, as a nonspecific neurologic manifestation. The findings in one of the conducted studies refer to amplitude of background <20 µV at 93% of “acute EEG,” versus only 21.4% of “follow-up EEG”, which was ‘caught’ when the average voltage went from 12.33  ±  5.09 µV in the acute EEGs to 32.8  ±  20.13 µV in the follow-up EEGs. Moreover, a total of 60% of acute EEGs showed intermittent focal rhythmic activity and there was no statistical significance in the correlation between voltage of acute EEG and clinical status, which had no relation to neurological pathological condition, including respiratory conditions that corelate to the EEG findings. Additionally, it was reported that the most common finding in COVID-19 was a nonspecific diffuse slowing EEG pattern. Highly distinctive low voltage EEG was noted, describing the low prevalence of epileptic activity with highly specific hypoxic mechanisms [131]. Another EEG-conducted study observed generalized background slowing in all patients and generalized epileptiform discharges with triphasic morphology in three patients, with focal electrographic seizures observed in one patient with a history of focal epilepsy and in another patient with no such history. Not that it is a novelty fact, but we can declare that pre-existing epilepsy can be a potential risk factor for COVID-19-associated neurological manifestations. Also five of eight patients who underwent EEG experienced a fatal outcome of infection in this investigation [132]. In the literature, there are large numbers of meta-analyses, reviews and case reports on neurological involvement in patients with COVID-19. Helms et al. found that the mean age of COVID-19 patients with neurological symptoms who were hospitalized in the intensive care unit (ICU) was 62 years, and that 75% of them were male. Additionally, it was reported that 82% of the patients hospitalized in the ICU had neurological manifestations, with the most common symptom being delirium [133]. It was noted in many investigations conducted on patients hospitalized in the ICU that the most common comorbidities are cardiovascular diseases and respiratory diseases among patients with neurological symptoms. Additionally, a study conducted showed that the most common chronic diseases among COVID-19 patients were hypertension (51.6%), previous ischemic stroke (37%), coronary artery disease (37%) and dementia (27.9%). In the study by Mao et al., it was reported that the rate of developing neurological symptoms was higher among patients with severe disease, as in our study data [134]. Headache and dizziness are common among COVID-19 patients with a frequency of headache in COVID-19 between 3 and 12.1% in one study, while 27% in another study. Dizziness was observed in 8% of COVID-19 patients [135]. Additionally, nine patients (5.8%) with a resistant headache and nine patients (5.8%) with persistent dizziness had consultations with the neurology department [136]. A very small number of retrospective studies have reported seizures in COVID-19 patients, with incidence ranging from 0.5 to 1.4%. All types of seizures have been reported in COVID-19 patients [137]. 

The most common radiological findings were cerebrocerebellar atrophy and diffuse ischemic gliotic areas, which were detected in 40.3% of MRI and 50.4% of CT examinations in a study conducted by Helmes et al. and Kandemir et al., who reported that the most common neuroimaging finding in COVID-19 patients was bilateral signal changes in FLAIR in MRI. Electromiography was performed on eight patients. Acute-onset ascending paraparesis was observed in one patient, tetraparesis in two patients and hypoesthesia in five patients. Although the rates of ischemic cerebrovascular disease (CVD) in COVID-19 patients are variable, researchers generally detected lower rates compared with the literature, perhaps due to the fact that the anticoagulant treatment was started as soon as the disease was detected in the patients with COVID-19 pneumonia. Thrombocytosis increased the risk of ischemia, which was expected, and high D-dimer, fibrinogen and CRP levels increased the risk of stroke in patients with concurrent COVID-19 and acute ischemic CVD. In a cohort study conducted, 96 patients who experienced vascular events associated with proinflammatory coagulopathy were found to have high CRP, D-dimer and ferritin levels. This situation was thought to have been due to endothelial dysfunction [138,139].

### 5.3. ML Application, Significance of the Issue

The use of AI is becoming more and more common. The pandemic has resulted in a lot of solutions that are beginning to be implemented in medical facilities. In 2022, ML applied at work [140] presented a new approach as a possible approach to gain deeper insights into the genetic information derived from target sequencing, to identify recurrent genetic patterns and improve the understanding of complex diseases. It is also worth presenting an example of the application of deep learning in healthcare. It is concerned with predicting the outcome of a heart transplant. The analyzers used data from the UNOS (United Network for Organ Sharing) register. They included over 27,000 patient records (2009–2011). Additionally, in this case, the deep learning model turned out to be a better tool for predicting short-term mortality than the aforementioned logistic regression. The results of this model have been made public in the form of an online tool that can be used to match recipient and donor [141]. The machine learning approach is at the top of the list of the research priorities related to the COVID-19 pandemic. The clinical complexity of COVID-19 ranges from asymptomatic cases to severe pneumonia [142], whose progression to respiratory failure is difficult to predict with a high degree of uncertainty both in the progression of the patient’s health status and in the speed at which patients develop respiratory failure requiring mechanical ventilation [143,144]. There is a need to create the ML model which would show a potential to produce predictive patterns that can be applied to assist and improve clinical decisions for a broad variety of outcomes [145,146]. One was created and used in response to the COVID-19 emergency [147,148]. A statistical learning model was created to assist clinicians in forecasting patients with COVID-19 who develop respiratory failure requiring mechanical ventilation with a reliable 48 h prediction of moderate to severe respiratory failure, with an accuracy of 84% that minimizes the FN rate. The level of performance of the model is in line with other ML tools used in different areas of medicine [149,150] and it is very useful in the COVID-19 clinical context where disease progression remains unpredictable both in the early virologic and in the late inflammatory phase. There are different models of machine learning constructed to follow a clinically oriented variables choice in acute respiratory infection. One of the first models was based on 31 variables that were collected from signs and symptoms assuming suboptimal prediction accuracy. Adding biomarkers including respiratory variables significantly increased the forecasting and predicting capacity of the model, but the best performance was observed in the boosted mixed model with approximately 20 variables, which from the clinician’s perspective may be difficult to obtain in routine hospital practice. What our approach offers in support to the decision-making process is a simple interpretation of the predictions. Moderate to severe respiratory failure was chosen as an outcome, being the most relevant time point in the natural history of severe COVID-19 pneumonia. At a clinical level, it represents the so-called “respiratory crush”, which marks acute lung injury and leads to mechanical ventilation in the ICU. At a public health level, this machine learning model might be helpful in optimizing scarce resources like ventilators and ICU beds. A few clinical risk scores have been developed and validated to predict the occurrence of critical illness in hospitalized patients with COVID-19, which were used at the time of the patient’s admission and included either the neutrophil/lymphocyte ratio or ten clinical variables including radiological diagnostic findings in order to predict critical illness and its possible outcome using a traditional statistical approach to generate a prediction algorithm [151,152]. There was also a ML model that used only three biomarkers in patients with COVID-19 [153]. It is important to add that COVID-19 does not affect only the respiratory system, but also the liver, kidneys, stomach, heart and central nervous system, defining it as multisystemic and therefore the limited number of parameters may not be sufficient to predict worsening in these patients. In order to investigate cognitive, EEG and MRI features in COVID-19 survivors up to 10 months after hospital discharge, the investigators used brain MRI at baseline. By using eLORETA, regional EEG densities and linear lagged connectivity were calculated and total brain and white matter hyperintensities were measured. The results showed that COVID-19 patients exhibited interrelated cognitive, EEG and MRI abnormalities 2 months after hospital discharge and cognitive and EEG findings improved at 10 months. Additionally, dysgeusia and hyposmia during acute COVID-19 were related with increased vulnerability in memory functions over time [154]. There is presumable impact of the lung–brain axis in the development of COVID-19 pneumonia and associated respiratory failure, highlighting clinical, neurophysiological and neuropathological evidence of SARS-CoV-2 neurotropism. The crucial role of the lung–brain axis explains the sensory inputs from the respiratory tract, which project to the central nervous system through cranial nerves, which carry special sensations from the nasal cavity through the cranial olfactory nerve. Additionally, the trigeminal nerve transmits the somatic sensations from the upper respiratory mucosa, large airways by glossopharyngeal nerve and the lungs by the vagus nerve. The glossopharyngeal nerve transports inputs from the carotid bulb, which is crucial for gas exchange and breathing regulation. Afferent signals from there project on the nucleus of the tractus solitarius in the pontomedullary region of the brainstem, allowing close monitoring of the respiratory function and potential noxious stimuli. Anatomically spoken, some neuronal populations related to the tractus solitaries belong to the dorsal respiratory group, which correspond to peripheral chemoreceptors, lung mechanisms, tissue damage and additional alteration of the dorsal respiratory group function comes through the pontine respiratory cell group and higher cortical structures. The dorsal respiratory group transmits impulses to the ventral respiratory group and the Pre-Bötzinger Complex and is responsible for the spontaneous rhythmic pattern of respiration [61]. Neurotropism of SARS-CoV-2 to the brainstem, where the vital functions are situated, followed by the constant neuroinflammatory innate response is already described with the involvement of the olfactory bulb and the pontomedullary region. All this can be defined as anatomical substrate and clinically correlates or predicts the disease severity and survival. Furthermore, prolonged damage may affect the recovery of COVID-19 patients, leading to persistent symptoms and eventually low quality of life. Further investigations about the role of the brainstem in COVID-19 are needed to improve diagnostic assessment and prompt research for new therapeutic strategies. The situation about SARS-CoV-2 and its related disease, correlated with brain damage and pulmonary disease, has posed a huge threat to the global population with millions of deaths and the creation of enormous social and healthcare pressure, therefore further investigations about consequences should be performed.

### 5.4. Dynamic Development of Techniques and Today’s Challenges

For distinguishing the infection of COVID-19 from non-COVID-19 groups, there are ten commonly used well-known convolutional neural networks: AlexNet, VGG-16, VGG-19, SqueezeNet, Xception, MobileNet-V2, ResNet-18, GoogleNet, ResNet-50 and ResNet-101.

The databases and algorithms that have been used to create better and more accurate algorithms are summarized in Table 3. Most of the works found are based on exactly the same data, and the authors have not seen the databases growing significantly in the last quarter. Depending on the type of data, their quality and quantity, one can distinguish between the techniques most commonly used in MRI, CT and X-ray. For COVID-19 classification from CT images, there is a conventional neural network [155], which is the current state-of-the-art for image classification [156,157]. In routine clinical practice using CT images, all those mentioned have achieved good performance, considering ResNet-101 could distinguish COVID-19 from non-COVID-19 cases with an AUC of 0.994 (sensitivity, 100%; specificity, 99.02%; accuracy, 99.51%) [158]. 

Vaid et al. showed how important access to open databases is. Using COVID-19 image data collection [26,159], they developed a deep learning model architecture based on a VGG-19 classifier [102]. Their COVID-19 detection model offers a very high accuracy of 96.3%. Nayak et al., by using COVID-19 data image collection [26] and NIH Chest X-rays dataset [15] databases, evaluated the effectiveness of eight pre-trained convolutional neural network models such as AlexNet, VGG-16, GoogleNet, MobileNet-V2, SqueezeNet, ResNet-34, ResNet-50 and Inception-V3 for the classification of COVID-19 from normal cases [103]. The best performance was obtained by ResNet-34 with an accuracy of 98.33%. Wang et al. used four different open databases: COVID-19 Image Data Collection [18], COVID-19 Chest X-ray Dataset, Actualmed COVID-19 Chest X-ray Dataset Initiative [16], COVID-19 Radiography Database [27] and RSNA Pneumonia Detection Challenge [28] to create COVID-Net, a deep convolutional neural network design, tailored for the detection of COVID-19 cases from chest X-ray images, which is open source and available to the general public [101]. COVID-Net achieved a very good accuracy of 93.3%. All of the above confirm the importance of creating open databases and free tools in the development of AI technology. It would not be possible if not for the common effort of many groups of scientists, who put a lot of time into the preparation and proper validation of image data, and were ready to share their results, which certainly contributed to the support of healthcare in the severe period of the COVID-19 pandemic.

Neural networks are used in many fields such as neuroimaging, medicine, biology, medical informatics or biomedical engineering and the latest ones focus on connections, brain–lung, lung–heart and heart–brain correlations [60,156,158]. Loey et al. [160], to identify NCOV-19 in chest X-ray images, have introduced a generative adversarial networks (GAN)-related deep transfer learning model. GAN was presented because of the robustness of the projected technique, which has been used for the screening of drugs with the help of text data [161]. The development of GAN was fast and impressive [162,163], resulting in an accuracy of 82.91%, and adequate performance in ResNet50. Muhammad et al. [164] presented a combined CNN-BiLSTM and experimental results, which demonstrate state-of-the-art focus on performance on three COVID-19 databases. Algorithm development in the period since the pandemic began has been very dynamic. It is not clear exactly when it was initiated that there are brain–lung correlations [165,166,167]. As soon as the first reports appeared, further algorithms occurred, allowing to expand the diagnosis, prediction, selection and presentation of the correlations in question. Unfortunately, at this point, the authors have not identified a database, with open access, that deals directly with brain–lung correlations [52]. From a technical point of view, despite the latest literature and the most up-to-date results, the authors are unable to reproduce most of the experiments because the databases have not been made available, or there is no open access to them. It should be noted that there is a great need to publish more databases, or parts of the data, which would accelerate the development of existing ready-made solutions.

While there is no doubt that AI is an important aspect of diagnostic imaging, you will also find expert voices, such as Alexander Selvikvåg Lundervold of the University of Applied Sciences of Western Norway in Bergen, Norway, who say that imaging is not the right way to go. First, physical signs of disease may not show up in scans until some time after infection, making it not very useful as an early diagnosis. Second, there is still not enough training data available, so it is difficult to assess the accuracy of such studies. Most image recognition systems—including those trained on medical images—are adapted to models first trained on ImageNet.

## 6. Summary and Conclusions 

Access to open databases appears to be a priority at this time. The review identified more than a dozen databases concerning lung diseases, COVID-19 and brain damage that provide a good basis for working on new or existing algorithms. Databases containing descriptions and X-ray, CT or MRI images of patients during and immediately after “COVID-19” are already relatively well described, and available to users. 

There are not many EEG databases that contain specific pathologies or diseases, in particular, considering tests of patients after COVID-19, who experience various neurological problems and require rehabilitation.

Some researchers presented findings showing a strong correlation between brain-related abnormalities and COVID-19, but future larger datasets with imaging-pathologic correlation may help better in understanding the common mechanisms of brain and lung injury and existing correlations.

To sum up, the use of machine learning forecasting algorithms increases the chances of, among others, early detection and diagnosis of diseases and threats to patients’ health or supports making clinical decisions and planning preventive measures. Such activities allow patients to maintain their physical and intellectual fitness for longer and improve their quality of life. When comparing the costs of prophylaxis and the costs of restorative medicine, it is worth looking for good ML solutions, the widespread use of which may also have a beneficial effect on the financial condition of the healthcare system. The analysis of the examples cited shows that using the full potential of ML depends largely on the size of the historical data sets, their updating, as well as the quality of the data they contain.

Nowadays, solutions can represent a possible starting point of a predictive tool for personalized medicine and advanced diagnostic applications. The results obtained by algorithms based on artificial intelligence show that it can improve the process of diagnosing patients, mainly thanks to supplementing the knowledge and experience of doctors.

## Figures and Tables

**Table 1 sensors-22-06312-t001:** Articles included in the review associated with lung diseases.

Study	Data	Disease	Algorithms Applied	Outcome Presentation
Bharati et al., 2020 [95]	NIH Chest X-rays dataset [15]	Different pulmonary diseases	VDSNet	F0.5 score of 68% with 73% validation accuracy.
Varschni et al., 2019 [96]	NIH Chest X-rays dataset [15]	Pneumonia detection	CNN (DenseNet-169)	Providing the dominating pre-trained CNN model and classifier.
Tang et al., 2020 [97]	NIH Chest X-rays dataset [15]	Pneumonia detection (detect pathology localization)	CNNs (AlexNet, VGGNet, ResNet, Inception-v3 (GoogLeNet), and DenseNet)	All CNNs have AUCs >0.96.
Annarumma et al., 2019 [98]	NIH Chest X-rays dataset [15]	Predict the priority level (i.e., critical, urgent, nonurgent, and normal). Pneumonia, Fibrosis, Mass	CNN (DenseNet)	Sensitivity 65%. Specificity 94%. AUC 0.609
Baltruschat et al., 2019 [99]	NIH Chest X-ray14 dataset [15]	Multi-label lung’s pathology classification	CNN (ResNet-50) and data acquisiton	AUC 0.822.
Bassi et al., 2020 [100]	NIH Chest X-rays dataset [15], COVID-19 Image Data Collection [18], CheXpert database [21]	Classification X-ray images as COVID-19, pneumonia and normal	DNN (DenseNet) and output neuron keeping	Test accuracies of 100%.
Wang et al., 2020 [101]	COVID-19 Image Data Collection [18], COVID-19 Chest X-ray Dataset, Actualmed COVID-19 Chest X-ray Dataset Initiative [16], COVID-19 Radiography Database [27], RSNA Pneumonia Detection Challenge [28]	COVID-19	COVID-Net	Test accuracy 93.3%. Sensitivity for COVID-19 cases: 91.0%.
Vaid et al., 2020 [102]	COVID-19 image data collection [30], NIH Chest X-rays dataset [15]	COVID-19	CNN (VGG-19)	Very high accuracy of 96.3%.
Nayak et al., 2021 [103]	COVID-19 data image collection [26], NIH Chest X-rays dataset [15]	COVID-19 detection	Different CNN models (VGG-16, Inception-V3, ResNet-34, MobileNetV2, AlexNet, GoogleNet, ResNet-50, and SqueezeNet)	Accuracy of 98.33% (for ResNet-34).
Chakravarthy et al., 2019 [104]	Lung Image Database Consortium (LIDC) [32]	Lung cancer detection	PNN	Classification accuracy of 90%.

**Table 2 sensors-22-06312-t002:** Articles included in the review associated with brain damage.

Study	Disease	Form	Complications/Manifestations
Young et al., 2020 [1]	Creutzfeldt–Jakob disease	Case report	Neurologic status progressed to mutism, right hemiplegia, spontaneous multifocal myoclonus, somnolence and agitation. He died 2 months after symptom onset.
Pimentel et al., 2022 [106]	Creutzfeldt–Jakob Disease, Rapidly Progressive Alzheimer’s Disease, and Frontotemporal Dementia	Report of Three Cases	Probable sporadic CJD. He died of sepsis, secondary to bacterial pneumonia 4 months after the symptom onset.
Pellinen et al., 2020 [107]	Remote ischemic stroke, epilepsy, brain disorders	Research electronic data capture [108]	In the absence of prior epilepsy or brain injury, seizures were rare.
Canham et al., 2020 [109]	Epilepsy, stroke	10 cases	The presence of focal disturbances or irritative abnormalities.
Louis et al., 2020 [110]	Epilepsy, stroke	22 cases	COVID-19-positive patients who were encephalopathic had a variety of epileptiform abnormalities on EEG.
Pastor et al., 2020 [111]	Stroke	20 cases	Some severely affected COVID-19 patients develop an encephalopathy with specific EEG features, with spectral and connectivity alterations, and raw tracings appear nearly physiological.
Ciolac et al., 2021 [112]	Creutzfeldt–Jakob Disease	Case report	The case of an elderly female patient with sporadic CJD that exhibited clinical deterioration with the emergence of seizures and radiological neurodegenerative progression following an infection with SARS-CoV-2 and severe COVID-19.
Galanopoulou et al., 2020 [2]	Epilepsy, Other neurological disorders	26 Ceribell EEGs, 4 routine and 7continuous EEG studies	Among COVID-19-positive vs. COVID-19-negative patients, respectively, were new onset encephalopathy (68.2% vs. 33.3%) and seizure-like events (14/22, 63.6%; 2/6, 33.3%), even among patients without prior history of seizures (11/17, 64.7%; 2/6, 33.3%). Sporadic epileptiform discharges (EDs) were present in 40.9% of COVID-19-positive and 16.7% of COVID-19-negative patients.
Petrescu et al., 2020 [113]	Stroke, Epilepsy	Patients with positive PCR for SARS-CoV-2 between 25 March 2020 and 6 May 2020 in the University Hospital of Bicêtre, 36 COVID-19 patients	The main indications were confusion or fluctuating alertness for 23 (57.5%) and delayed awakening after stopping sedation in ICU in six (15%). EEGs were normal to mildly altered in 23 (57.5%) contrary to the 42.5% where EEG alterations were moderate in four (10%), severe in eight (20%) and critical in five (12.5%).
Kubota et al., 2021 [114]	Epilepsy Encephalopathy	12 studies with 308 patients fulfilled the eligibility criteria for inclusion in the meta-analysis	The proportion of abnormal background activity in patients with COVID-19 was high (96.1%).
Antony and Haneef, 2020 [115]	Encephalopathy, Epilepsy	Available data was analyzed from 617 patients with EEG findings reported in 84 studies.	Frontal findings are frequent and have been proposed as a biomarker for COVID-19 encephalopathy.
Roberto et al., 2020 [116]	COVID-19 patients	177 COVID-19 patients	COVID-19 patients may frequently manifest with abnormal EEG particularly in severe cases.

**Table 3 sensors-22-06312-t003:** Algorithms used in the process of dynamic development of AI techniques.

Study	Data	Algorithm Applied
Liang et al., 2020 [151]	Retrospective cohort of patients with COVID-19 from 575 hospitals	Estimate of the risk that a hospitalized patient with COVID-19 will develop critical illness
Vente et al., 2022 [156]	COVID-19 (iCTCF) dataset, e.g., 4001 positive CT, 9979 negative CT [168,169,170]	Comparing the performance of a variety of popular 2D and 3D CNN architectures
Ali Abbasian Ardakani et al., 2020 [171]	1020 CT slices from 108 patients with laboratory-proven COVID-19 (the COVID-19 group) and 86 patients with other atypical and viral pneumonia diseases (the non-COVID-19 group) were included [172]	Were used to distinguish infection of COVID-19 from non-COVID-19 groups: AlexNet, VGG-16, VGG-19, SqueezeNet, GoogleNet, MobileNet-V2, ResNet-18, ResNet-50, ResNet-101 and Xception
Nayak et al., 2021 [103]	COVID-19 data image collection [26] and NIH Chest X-rays dataset [15] databases	Evaluating the effectiveness of eight pre-trained convolutional neural network models such as AlexNet [173], VGG-16, GoogleNet, MobileNet-V2, SqueezeNet, Res-Net-34, ResNet-50 and Inception-V3
Wang, 2020 [101]	COVID-19 Image Data Collection [174], COVID-19 Chest X-ray Dataset, Actualmed COVID-19 Chest X-ray Dataset Initiative [175], COVID-19 Radiography Database [176] and RSNA Pneumonia Detection Challenge [177]	Creating COVID-Net, a deep convolutional neural network design
Loey et al., 2020 [160]	742 CT images [178,179,180,181]	Introduced a generative adversarial networks (GAN)-related deep transfer learning model
Muhammad et al., 2022 [164]	500 no-findings and 500 pneumonia class frontal chest X-ray images [179,180,181,182,183,184,185]	Presented a combined CNN-BiLSTM
Ucar et al., 2020 [186]	5232 chest X-ray images from children [101]	COVIDiagnosis-Net Bayes-SqueezeNet
Apostolopoulos et al., 2020 [187]	A collection of X-ray images from Cohen (1427 X-ray images) [159,177,188,189]	The pretrained CNNs
Li and Zhu 2020 [190]	Chest X-ray8 dataset (108,948 lung disease cases) [177]	DenseNet
Wang and Wong 2020 [101]	13,975 CXR images across 13,870 patient cases [21,26,27,176,177,191,192]	Tailored CNN
Chowdhury et al., 2020 [192]	Muhammed [164] SIRM COVID-19 database [193] Novel Corona Virus 2019 Dataset [18] COVID-19 Chest imaging at thread reader C. Imaging, This is a Thread of COVID-19 CXR (All SARS-CoV-2 PCR+) From my Hospital (Spain). I Hope it Could Help [194] RSNA-Pneumonia-Detection-Challenge [105] Chest X-ray Images (pneumonia): [174,195]	Sg-SqueezeNet
Ozturk et al., 2020 [174]	127 X-ray images [159,188]	DarkCovidNet

## Data Availability

No new data were created or analyzed in this study. Data sharing is not applicable to this article.

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
