# Peer review of "Database and AI Diagnostic Tools Improve Understanding of Lung Damage, Correlation of Pulmonary Disease and Brain Damage in COVID-19"

_sensors, 2022, doi:10.3390/s22166312_

Round 1

Reviewer 1 Report

Overall, this is an interesting topic, and the tables nicely summarized previous papers using AI algorithms for detecting lung diseases, brain diseases, and Covid patients. 

Currently, this paper has a few limitations: 

1. The title and the abstract aims to suggest how AI algorithms can be used to understand the correlation between pulmonary diseases and brain diseases in COVID patients. However, the majority of the paper actually focused on AI algorithms applied in each disease category separately, and there were only a few studies covering both diseases or COVID patients. There was no in-depth discussion on how AI algorithms focused on each disease led to other studies, that eventually helped readers understand the correlation.

For example, if an article tries to say we can use AI algorithms to understand the correlation between admission rate and student sport performance, yet most studies either focused on predicting admission rates with no sport data or predicting sport performance without admission information, while only providing 1 or 2 papers really looking at both. This raises a question on the point of bringing up these other papers unrelated or unhelpful to understanding the correlation. 

Similarly, the writing of this paper leads me to wonder how these previous work focusing on lung disease or brain disease helped understand the correlation in COVID patients. 

2. If the authors listed these papers to acknowledge the development of these core algorithms, these AI tools mostly benefit from the progress in computer vision in general, which further led me to wonder if the authors should cite the original papers developing these core algorithms (ResNet, etc) and discuss how these progress lead to helping understand COVID diseases instead. 

3. Many algorithms listed in this review were not state-of-the-art (SOTA) anymore. Newer SOTA algorithms focused on transformer architectures, and there were studies using such architecture in predicting COVID.  

 I would recommend a major revision to improve the writing of the paper to be more organized and cohesive. 

Author Response

Thank you for all your comments. We tried to include all of them in the revised version of the manuscript. All changes are visible in "track changes" mode.

  1. The title and abstract aim to suggest how AI algorithms can be used to understand the correlation between pulmonary diseases and brain diseases in COVID patients. However the majority of the paper actually focused on AI algorithms applied in each disease category separately, and there were only a few studies covering both diseases or COVID patients. There was no in-depth discussion on how AI algorithms focused on each disease led to other studies, that eventually helped readers understand the correlation.

Answer:  In response to a comment, the literature has been expanded to include current items.  We focused mainly on subsection 3.2 taking into account the reviewers' suggestions. In fact, we find in our article that AI algorithms were applied in each disease category separately, and there were only a few studies covering both diseases or COVID patients. Our intention was to identify open-access databases, not only those with COVID-19 but other lung diseases that can be used for further analysis. It seems to us that "how AI algorithms focused on each disease led to other studies, that eventually helped readers understand the correlation" could be a topic for another short article. The considerations arise from the fact that this is a very broad topic.

The writing of this paper leads me to wonder how these previous works focusing on lung disease or brain disease helped understand the correlation in COVID patients.

Answer:  The topic is not yet 100% exhausted. As we tried to present in the paper, the literature we selected, particularly that of 2022, shows that this phenomenon is not yet 100% clear and confirmed.

Our purpose was to highlight the emerging correlation between pulmonary disease and brain damage and gather all open databases and tools related with those in one article, to help scientists, interested in AI algorithms, find all the data needed to conduct their own research in one place. As the topic seems to be new, but important in the face of post COVID syndrome and coming the next waves of contagion.

  1. If the authors listed these papers to acknowledge the development of these core algorithms, these AI tools mostly benefit from the progress in computer vision in general, which further lead me to wonder if the authors should cite the original papers developing these core algorithms (ResNet, etc.) and discuss how these progress lead to helping understand COVID diseases instead.

Answer: Thank you for your suggestion. We used ready made descriptions of algorithms that we found in literature from 2019 until may 2022. The nature of the title correlation is in constantly discussion, which is described more broadly in paragraph 5.2 in our Discussion. When completing our article, we added a paragraph 5.4 in Discussion, in which appeared some of the original works developing core algorithms (such as ResNet-101).

  1. Many algorithms listed in this review were not state-of-the-art (SOTA) anymore. Newer SOTA algorithms focused on transformer architectures and there were studies using such architecture in predicting COVID.

Answer: In Table 1. we added the specific types of applied algorithms in all of the mentioned works.  The types of use, algorithms, specific convolutional neural networks, are in my knowledge, state-of-the-art for image classification, for example: AlexNet, VGGNet, GoogleNet, ResNet-50. We also added the paragraph 5.4 in the Discussion, in which we describe, taken from the literature, examples of algorithms accepted as SOTA for automatic COVID-19 classification. We have added several items to our work, these are also review with descriptions of algorithms SOTA.

  1. I would recommend a major revision to improve the writing of the paper to be more organized and cohesive.

Answer: We improved, and reorganized some parts of our work considering all 3 reviews. Adhering to the comments, we think the readability of the work has improved.

Reviewer 2 Report

1. All Tables lack the legned. Please add legend on the Table.

2. Please alignment words in Complications/Manifestations in Table. 

Author Response

Thank you for your comments. We tried to include all of them in the revised version of the manuscript. All changes are visible in "track changes" mode.

  1. Add legends to tables.

Answer: Headers added.

  1. Alignment words in Complications/Manifestations in Table.

Answer: The alignment was adjusted.

Reviewer 3 Report

This is a review-type article for the database and AI techniques for the correlation of pulmonary disease and brain damage in COVID. A major revision is needed before the recommendation of acceptance.

Comment 1. Title, Clarify the correlation between pulmonary disease and ...
Comment 2. Abstract:
(a) Refer to the journal’s template. Organize the abstract in single paragraph. Shorten the contents to ensure it is below the maximum word count.
(b) Clarify “period 2019-2022” for the ending month of 2022 for the consideration.
Comment 3. Keywords, more terms should be included to better reflect the scopes of the paper.
Comment 4. Section 1 Introduction:
(a) Clarify “period 2019-2022”.
(b) Update “PACS (Picture Archiving and Communication Systems)” as “picture archiving and communication systems (PACS)”. Apply the same for other acronyms where necessary.
(c) Elaborate the importance of the research topic and if there was related works (review articles in the topic).
(d) Summarize the research contributions of the paper.
Comment 5. Update the list of references. Refer to the journal’s template.
Comment 6. Section 2 Machine Learning Tools:
(a) The contents before Subsection 2.1 Open Databases should be categorized as a new subsection.
(b) Subsection 2.1 should be rewritten to focus on COVID-19 datasets to align with the title of the paper.
(c) Formal citations are required for the URLs.
Comment 7. Section 3 Systems and Tools:
(a) Update the section’s heading where “Tools”s are used in both Sections 2 and 3.
(b) Ensure proper spacing between the wordings and in-text citations [x].
Comment 8. Section 4 Results:
(a) Confirm the symbols for the double quotations “”
(b) It seems that COVID-19 is missing.
(c) Please share the codes for the search where it seems to be incomplete search with only 10 articles reported in Table 1.
(d) Headings of Tables are missing.
(e) Some works in Tables 1 and 2 were not COVID-19 related.
Comment 9. Section 5 Discussion:
(a) The discussion is not well organized. Consider to group relevant contents into same paragraph.
(b) The discussion should be supported by the analysis of the literature review.
(c) Technical discussion should be made on the AI techniques.
Comment 10. The conclusion of the paper should be well justified.

Author Response

Thank you for all your comments. We tried to include all of them in the revised version of the manuscript.  All changes are visible in "track changes" mode.

Comment 1. Title, clarify the correlation between pulmonary disease and...

Answer: We made changes to the work to make the topic more understandable.

Comment 2. Abstract:

  • Refer to a journal’s template. Organize the abstract in a single paragraph. Shorten the contents to ensure it is below the maximum word count.

Answer: We have corrected the abstract according to the current scheme.

  • Clarify the period 2019-2022 for the ending month of 2022 for consideration.

Answer: We have improved the range. 05.2022 to be precise.

Comment 3. Keywords and more terms should be included to better reflect the scope of the paper.

Answer: We changed the words, and added new ones. CNS has been removed and replaced.

Comment 4. Section 1 Introduction:

  • Clarify period 2019-2022.

Answer: Period clarified to May 2022.

  • Update PACS as picture archiving and communication systems (PACS). Apply the same for other acronyms where necessary.

Answer:  Updated.

  • Elaborate on the importance of the research topic and if there were related works (review articles in the topic).

Answer: We added the literature that confirms existing of a correlation between COVID and brain damage and explore that topic.

  • Summarize the research contributions of the paper.

Answer: Our purpose was to highlight the emerging correlation between pulmonary disease and brain damage and gather all open databases and tools related to those in one article, to help scientists, interested in AI algorithms, find all the data needed to conduct their own research in one place. As the topic seems to be new, but important in the face of post COVID syndrome and the next new waves of contagion.

Comment 5. Update the list of references. Refer to the journal’s template.

Answer: Updated.

Comment 6. Section 2 Machine Learning.

  • The contents before Subsection 2.1 Open databases should be categorized as a new subsection.

Answer: New subsection added: 2.1 Tools, libraries, and blogs.

  • Subsection 2.1 should be rewritten to focus on COVID-19 datasets to align with the title of the paper.

Answer: Some databases are not directly related to COVID 19 but, for example, relate to various other lung diseases, which are also used in the machine learning process, in the topic under discussion, which can be using in the topic we are presenting, in the context of correlation. Our intention was not to gather only and exclusively COVID-19 but also others that relate to lung diseases.

The objective of this paper is to perform a systematic review, to summarize the electroencephalogram (EEG) findings in patients with coronavirus disease (COVID-19) and databases and tools used in artificial intelligence algorithms, supporting the diagnosis and correlation between lung disease and brain damage.

Formal citations are required for the URLs.

Answer: We formatted and added URL’s for formal citations.

Comment 7. Section 3 Systems and Tools

  • Update the section’s heading where Tools are used both in Sections 2 and 3.

Answer: Changed Section 3 header to “Systems and Applications.”

  • Ensure proper spacing between the wordings and in-text citations [x].

Answer: Checked and changed where necessary.

Comment 8. Section 4 Results

  • Confirm the symbol for the double quotations “”.

Answer: Symbols changed to ‘’’ where necessary.

  • It seems that COVID-19 is missing.

Answer: Added “COVID-19” in search terms.

  • Please share the codes for the search where it seems to be incomplete search with only 10 articles reported in Table 1.

Answer: We chose only those 10 articles because of the related databases. We wanted to show the results of the AI algorithms used only on open databases, so we chose the articles with the best results outcome presentations which we were able to find.  

  • The heading of the Tables is missing.

Answer: Added headings.

  • Some works in Tables 1 and 2 were not Covid-19 related.

Answer: Some are not directly related to COVID 19 but, for example, relate to various other lung diseases, which are also used in the machine learning process, in the topic under discussion. Our intention was not to gather only and exclusively COVID-19 but also others that relate to lung diseases.

Comment 9. Section 5 Discussion

  • The discussion is not well organized. Consider to group relevant contents into the same paragraph.

Answer: Paragraph 5.2 and 5.4 are now one paragraph: 5.3 application ML, the significance of the issue.  The paragraph has been modified. Additional literature has been added.

  • The discussion should be supported by the analysis of the literature review.

Answer: Literature has been completed.  

  • Technical discussion should be made on the AI techniques.

Answer: We have supplemented the discussion with a short passage as suggested. In order not to duplicate the content most of it is in paragraphs 3 and 5.3, 5.5.  We have introduced an additional paragraph without describing the individual networks, since there are, of course, many descriptions in the literature.

Also taking into account the suggestion of other reviewers, we have added a paragraph that presents a discussion based on the latest algorithms, state of the art.

Comment 10. The conclusion of the paper should be well justified.

We have corrected, both the conclusions and the text.

Round 2

Reviewer 1 Report

Thanks authors for incorporating the suggestions. The writing structure improves and I would proceed with acceptance. 

Author Response

Thank you.

We still introduced a summary table, which contains additional bases that were used to develop "state of the art" algorithms.

Reviewer 3 Report

I have some minor follow-up comments:

Comment 1: Refer to a journal’s template. Organize the abstract in a single paragraph. Shorten the contents to ensure it is below the maximum word count.

Answer: We have corrected the abstract according to the current scheme.

Follow-up comment: It has not been updated for the organization in single paragraph.

Comment 2: Some works in Tables 1 and 2 were not Covid-19 related.

Answer: Some are not directly related to COVID 19 but, for example, relate to various other lung diseases, which are also used in the machine learning process, in the topic under discussion. Our intention was not to gather only and exclusively COVID-19 but also others that relate to lung diseases.

Follow-up comment: If other topics are included, the title and abstract should be updated.

We have corrected, both the conclusions and the text.

Comment 3: Technical discussion should be made on the AI techniques.

Answer: We have supplemented the discussion with a short passage as suggested. In order not to duplicate the content most of it is in paragraphs 3 and 5.3, 5.5. We have introduced an additional paragraph without describing the individual networks, since there are, of course, many descriptions in the literature.

Follow-up comment: More tables should be added instead of written descriptions for the summary of many works.

Author Response

We have made the changes. 

We have added a table summarizing the discussion. It still contains databases and links to publications with the algorithms used.

In the subject discussed, it seems to us that the topic is already sufficiently exhausted.

New bibliography items also appeared along with the table.